# *LAMB3* Missense Variant in Australian Shepherd Dogs with Junctional Epidermolysis Bullosa

**DOI:** 10.3390/genes11091055

**Published:** 2020-09-07

**Authors:** Sarah Kiener, Aurore Laprais, Elizabeth A. Mauldin, Vidhya Jagannathan, Thierry Olivry, Tosso Leeb

**Affiliations:** 1Institute of Genetics, Vetsuisse Faculty, University of Bern, 3001 Bern, Switzerland; sarah.kiener@vetsuisse.unibe.ch (S.K.); vidhya.jagannathan@vetsuisse.unibe.ch (V.J.); 2Dermfocus, University of Bern, 3001 Bern, Switzerland; 3The Ottawa Animal Emergency and Specialty Hospital, Ottawa, ON K1K 4C1, Canada; alaprais@oaesh.com; 4School of Veterinary Medicine, University of Pennsylvania, Philadelphia, PA 19104, USA; emauldin@vet.upenn.edu; 5Department of Clinical Sciences, College of Veterinary Medicine, North Carolina State University, Raleigh, NC 27607, USA

**Keywords:** dog, *Canis lupus familiaris*, whole genome sequence, wgs, dermatology, genodermatosis, skin, laminin, precision medicine

## Abstract

In a highly inbred Australian Shepherd litter, three of the five puppies developed widespread ulcers of the skin, footpads, and oral mucosa within the first weeks of life. Histopathological examinations demonstrated clefting of the epidermis from the underlying dermis within or just below the basement membrane, which led to a tentative diagnosis of junctional epidermolysis bullosa (JEB) with autosomal recessive inheritance. Endoscopy in one affected dog also demonstrated separation between the epithelium and underlying tissue in the gastrointestinal tract. As a result of the severity of the clinical signs, all three dogs had to be euthanized. We sequenced the genome of one affected puppy and compared the data to 73 control genomes. A search for private variants in 37 known candidate genes for skin fragility phenotypes revealed a single protein-changing variant, *LAMB3*:c.1174T>C, or p.Cys392Arg. The variant was predicted to change a conserved cysteine in the laminin β3 subunit of the heterotrimeric laminin-322, which mediates the binding of the epidermal basement membrane to the underlying dermis. Loss-of-function variants in the human *LAMB3* gene lead to recessive forms of JEB. We confirmed the expected co-segregation of the genotypes in the Australian Shepherd family. The mutant allele was homozygous in two genotyped cases and heterozygous in three non-affected close relatives. It was not found in 242 other controls from the Australian Shepherd breed, nor in more than 600 other controls. These data suggest that *LAMB3*:c.1174T>C represents the causative variant. To the best of our knowledge, this study represents the first report of a *LAMB3*-related JEB in domestic animals.

## 1. Introduction

When a human or animal, usually at or soon after birth, develops erosions and epithelial sloughing on the mucosae, areas of friction, and extremities, a genetic disorder of skin fragility is to be considered. A consensus reclassification of skin fragility disorders was published recently, which separates those that affect the basement membrane itself or the basal keratinocytes (i.e., hereditary epidermolysis bullosa (EB) variants) from others, in which the separation occurs more superficially in the epidermis [1]. In this reclassification, four main categories of inherited “classical” EB are proposed, which reflect the differences in the level of cleavage in the basement membrane zone [1]. Also included in this reclassification are four new categories of epidermal disorders of skin fragility associated with 20 possibly mutated genes, namely: peeling skin disorders, erosive skin fragility disorders, keratinopathic ichthyoses, and pachyonychia congenita [1]. Finally, a single syndromic connected tissue disorder with (dermal) skin fragility associated with *PLOD3* variants and a lysyl hydroxylase-3 deficiency was also included in this group of diseases [1]. All of the known 37 candidate genes for these human diseases are summarized in Table 1.

In domestic dogs, only two other epidermal disorders of skin fragility have been reported, namely: epidermolytic ichthyosis associated with a *KRT10* variant in Norfolk terriers [2], and ectodermal dysplasia/skin fragility syndrome with a *PKP1* variant in Chesapeake Bay Retrievers [3] (Appendix A). In contrast, cases of hereditary EB have been recognized for decades, and the causative genetic variants have now been characterized in three canine, one feline, two equine, two ovine, and five bovine EB variants (Appendix A).

In dogs, there is at least one example for each of the three main subtypes of classical EB in which the genetic defect has been reported, namley: a *PLEC* variant in the EB simplex of Eurasier dogs in the USA [4]; a *LAMA3* variant in the junctional EB of German Shorthaired Pointers in France [5]; and *COL7A1* variants in the dystrophic EB (mild) in Golden Retrievers, also in France [6], or severe in Central Asian Shepherds [7].

Laminin-332, a rod-like heterotrimer composed of the laminin α3, β3, and γ2 chains, is a critical component of hemidesmosomes, adhesion complexes that attach the basal epidermal keratinocytes to the underlying dermal connective tissue [8,9,10]. The prominent role of laminin-332 for skin integrity stems from its ability to link two important molecules—one in the epidermis and the other in the dermis. Via its carboxy-terminus, laminin α3 binds to the external domains of the integrin α6β4 that protrude from the basal keratinocytes. At the other end of the laminin trimer, the amino-terminal domains of the laminin β3 and γ2 chains bind to the NC1 amino-terminus of the superficial dermal collagen type VII [11].

Genetic variants in the *LAMA3*, *LAMB3,* and *LAMC2* genes that encode the laminin α3, β3, and γ2 chains are causative for the intermediate and severe forms of junctional EB (JEB), not only in humans [1], but also in animals (Appendix A). Variants in any one of these genes can lead to a similar phenotype, as the abnormal expression or function of either of the three individual laminin chains is expected to impair the assembly or the function of the entire laminin-332. A good example of this phenomenon is the near identical phenotype exhibited by American Saddlebred horses with severe JEB associated with a *LAMA3* variant [12], and that found in Belgian, Breton, Comtois, and Italian draft horses caused by a *LAMC2* variant [13,14,15].

While JEB subsets associated with *LAMB3* variants are common in humans [16,17,18], they have not yet been reported in animals. So far, an abnormal epidermal expression of laminin β3—without investigation of the underlying molecular genetics—has only been shown in a single cat exhibiting a phenotype of mild EB [19].

Herein, we report a missense variant in *LAMB3,* which we believe is causative of a JEB phenotype with intermediate severity in a litter of Australian Shepherds in Ontario, Canada. Of clinical interest is the demonstration, for the first time or so it seems, of intestinal epithelial sloughing in a case of animal JEB.

## 2. Materials and Methods

### 2.1. Ethics Statement

The affected Australian Shepherds in this study were privately owned, and skin and biopsy samples were collected with the consent of their owners. The collection of all other blood samples was approved by the “Cantonal Committee for Animal Experiments” (Canton of Bern; permits 75/16 and 71/19).

### 2.2. Animal Selection

This study included 247 Australian Shepherds. Genomic DNA was either isolated from EDTA blood samples with the Maxwell RSC Whole Blood Kit, or from formalin-fixed paraffin-embedded (FFPE) tissue samples with the Maxwell RSC DNA FFPE Kit using a Maxwell RSC instrument (Promega, Dübendorf, Switzerland).

### 2.3. Histopathological Examinations

Skin punch biopsies (8 mm) were obtained under general anesthesia. The samples were fixed in 10% neutral buffered formalin and routinely processed, including staining with hematoxylin and eosin.

### 2.4. Whole Genome Sequencing

An Illumina TruSeq PCR-free DNA library with ~400 bp insert size of a JEB affected Australian Shepherd was prepared. We collected 175 million 2 × 150 bp paired-end reads or 18.9× coverage on a NovaSeq 6000 instrument. The reads were mapped to the dog reference genome assembly CanFam3.1 and were aligned as previously described [20]. The sequence data were submitted to the European Nucleotide Archive, with study accession number PRJEB16012 and sample accession number SAMEA6862980.

### 2.5. Variant Calling

Variant calling was performed as previously described [20]. To predict the functional effects of the called variants, SnpEff software [21], together with NCBI annotation release 105 for the CanFam 3.1 genome reference assembly, was used. For variant filtering, we used 73 control genomes (Appendix A).

### 2.6. Gene Analysis

We used the dog reference genome assembly CanFam3.1 and NCBI annotation release 105. Numbering within the canine *LAMB3* gene corresponds to the NCBI RefSeq accession numbers XM_014115071.2 (mRNA) and XP_013970546.1 (protein). For a multiple species comparison of the *LAMB3* amino acid sequences, we used the following accessions: NP_000219.2 (*Homo sapiens*), NP_001075065.1 (*Bos taurus*), XP_023496552.1 (*Equus caballus*), NP_001264857.1 (*Mus musculus*), NP_001094311.1 (*Rattus norvegicus*), XP_425827.3 (*Gallus gallus*), XP_002933550.2 (*Xenopus tropicalis*), and XP_700808.6 (*Danio rerio*).

### 2.7. Sanger Sequencing

To confirm the candidate variant *LAMB3*:c.1174T>C, and to genotype all of the dogs in this study, Sanger Sequencing was used. A 403 bp PCR product was amplified from genomic DNA using AmpliTaqGold360Mastermix (Thermo Fisher Scientific, Waltham, MA, USA) and primers 5′-TCT TGT GCC AAG CAC TGT TC-3′ (Primer F) and 5′-GGC ATA GGT GAG TCC CGT AA-3′ (Primer R). A smaller PCR product of 153 bp size was amplified for FFPE-derived DNA with primers 5′-GGT GGC TGC TTT TCT GTC TC-3′ (Primer F) and 5′-GGT GAG TCC CGT AAA TCC TG-3′ (Primer R). After treatment with shrimp alkaline phosphatase and exonuclease I, PCR amplicons were sequenced on an ABI 3730 DNA Analyzer (Thermo Fisher Scientific). Sanger sequences were analyzed using the Sequencher 5.1 software (GeneCodes, Ann Arbor, MI, USA).

## 3. Results

### 3.1. Family Anamnesis, Clinical Examinations, and Histopathology

Three Australian Shepherd puppies with severe skin lesions were identified in a highly inbred litter resulting from a father–daughter mating. The litter consisted of three affected and two non-affected puppies that were born out of normal parents. The pedigree relationships were suggestive for a monogenic autosomal recessive inherited disease (Figure 1).

At the time of their first presentation to the breeder’s veterinarian for vaccination at 7 weeks of age, the three affected puppies were noted to have ulcers in the mouth, inner pinnae, and abdomen. The puppies reportedly also had marked lymph node enlargement. The average weight of the affected puppies was half that of their unaffected siblings.

At 17 weeks of age, one of the affected dogs, a blue merle with copper intact female was presented to the dermatologist for evaluation of severe ulceration of both the oral cavity and haired skin. Ulcers were located on the gingival and buccal mucosa, tongue, and hard and soft palates (Figure 2a). The concave pinnae, bilaterally, were also ulcerated, oozing, and covered with exudate (Figure 2b), but the otoscopic examination only revealed mild erythema in the ear canal. Several footpads, either digital or central, were also ulcerated (Figure 2c), and four claws were missing or misshapen. Erosions and ulcers were covered by thick crusts on the elbows, hocks, and the tip of the tail. The vulva and anus were grossly normal.

Thoracic auscultation, abdominal, and lymph node palpation were all unremarkable. Blood was collected for a complete blood count and a serum chemistry panel, and the most relevant changes were a mild regenerative anemia (hemoglobin: 129 (reference range: 134–207 g/L); reticulocytes: 118 (10–110 k/µL)) and hypoproteinemia (total proteins: 46 (55–75 g/L); albumin: 23 (27–39 g/L); globulins: 23 (24–40 g/L)). To determine if these abnormal changes were due to digestive ulcers, an upper gastrointestinal endoscopy was performed under general anesthesia, two weeks after the original admission to the specialty clinic. The esophagus appeared normal, and the stomach and duodenum were hyperrhemic but did not show a visible loss of epithelium; endoscopic biopsies were nevertheless collected from the stomach and duodenum. During this general anesthesia, punch skin biopsies were collected from the concave pinnae, footpads, and oral cavity (hard palate, buccal mucosa, and tongue).

Microscopically, the skin and mucosal biopsy samples all exhibited limited-to-widespread epidermal detachment (Figure 2d), and ulcers were covered with serocellular crusts; inflammation was sparse in non-ulcerated areas. In some sections (as in Figure 2d), the basement membrane could be discerned at the base of the clefts, thus suggesting the diagnosis of JEB. The endoscopic biopsies from the stomach (pyloric and nonpyloric areas) and duodenum all showed mild-to-moderate inflammation with lymphocytes, plasma cells, and eosinophils, with a detachment of the epithelium from the underlying lamina propria (Figure 2e). Because of the severity of the lesions, the dog was euthanized at 7.5 months of age.

The medical records of the two other affected puppies were also reviewed. A blue merle with copper intact male puppy was noted to have ulcers on the tongue, gingiva, soft palate, pharynx, tonsils, and larynx. Skin ulcers were found on the concave pinnae and pressure points of one elbow, one hock, and both stifles. Because of the worsening lesions, this puppy was euthanized at 16 weeks of age, with biopsy samples of the tongue, soft palate ear, and footpad collected post-mortem. As for the samples obtained from the littermate described above, microscopic lesions consisted of subepidermal vesicles leading to dermo-epidermal separation, ulceration, and granulation tissue.

The third affected puppy, a blue merle female, had been euthanized at 5 months of age because of severe gingival, labial, oropharyngeal, and esophageal ulceration. The dog had crusts on the chin, ulcers and crusts on the concave pinnae and footpads, and exudate at the base of multiple claws; samples for histopathology were not collected.

Finally, both the sire and dam, as well as the two healthy siblings, were examined by veterinarians, and they were deemed to be free of skin lesions.

### 3.2. Genetic Analysis

In order to characterize the underlying causative genetic variant, we sequenced the genome of one affected dog at 18.9× coverage and searched for homozygous variants in the 37 genes known to cause human skin fragility (Table 1), which were exclusively present in the affected dog and absent from the genomes of 73 other dogs (Table 2, Appendix A).

This analysis identified a single homozygous private protein-changing variant in *LAMB3*, a known candidate gene for JEB in humans [1]. The variant can be designated as Chr7:8,286,613A>G (CanFam3.1 assembly). It is a missense variant, XM_014115071.2:c.1174T>C, predicted to change a highly conserved cysteine residue in the third EGF-like domain of laminin β3, XP_013970546.1:p.(Cys392Arg).

We confirmed the presence of the *LAMB3* missense variant by Sanger sequencing (Figure 3). The mutant allele showed the expected co-segregation with JEB in the available family. The two available DNA samples from the JEB affected puppies carried the mutant allele in a homozygous state, while their parents were heterozygous, as expected for obligate carriers (Figure 1).

We determined the genotypes at *LAMB3*:c.1174T>C in a cohort comprising 247 Australian Shepherd dogs, including the index family. The mutant *LAMB3* allele was not detected in the homozygous state in any of the 245 non-affected Australian Shepherd dogs or 663 dogs from other breeds. Three of these dogs, all members of the index family, carried the mutant *LAMB3* allele in a heterozygous state (Table 3).

## 4. Discussion

In the affected Australian Shepherds described in this study, the age of lesion onset, as well as the presence of ulceration in the oral cavity and pressure points on the limbs with a loss of claws, all suggested the clinical diagnosis of a skin fragility disorder, of which EB is the most representative disease group in domestic animals and humans (Appendix A). Because of the resembling phenotypes, clinical signs cannot alone reliably permit differentiation between the three main subtypes of animal EB (simplex, junctional, and dystrophic). For a more precise diagnosis, the specific location of the dermo-epidermal separation must be determined, for example, with a periodic acid Schiff (PAS) stain to visualize the glycoproteins in the basement membrane lamina densa [7], single or double antigen immunomapping [23], or transmission electron microscopy [4]. In this case, the routine histopathology enabled the visualization of the basement membrane delineating the contour of dermal imprints of the epidermal ridges, thus establishing that clefting occurred in a supra-lamina densa manner; this confirmed the diagnosis of JEB.

There is only one other occurrence of JEB in the canine species [5,23,24]. In the early 1990s, JEB was first discovered in German Shorthaired Pointers in the French Alps. The clinical signs were indistinguishable from those present in the Australian Shepherds described herein. Both the Pointer and Australian Shepherd puppies exhibited the first clinical signs weeks after, and not at, birth. In both cases, lesions consisted of ulcerative skin lesions affecting the inner (medial and concave) pinnae, footpads, and at pressure points of the extremities [23,24]. Shedding of the claws was also reported [24]. Of interest is that dental enamel abnormalities, a common finding in human JEB [1], were not recognized in either the German Shorthaired Pointers or the Australian Shepherds described in this study. A unique finding seen in one of the three Australian Shepherd puppies was the endoscopic observation of duodenal hyperrhemia, which was found on the histopathology to be associated with an extensive detachment of the digestive epithelium from its underlying connective tissue. Unfortunately, as ulceration of the duodenum was not seen during endoscopy, we cannot rule out that the digestive epithelial detachment seen on the histopathology might have been artifactual. Nevertheless, the *LAMA3, LAMB3,* and *LAMC2* genes, which encode the three laminin-332 chains, are all expressed in the small intestine [25]. As a result, based on our hypothesis, that the *LAMB3* missense variant affects the adhesive function of the laminin-332, it is conceivable that any trauma to the small intestine during the endoscopic biopsy process could result in a forced epithelial separation from the lamina propria, a phenomenon that normally does not happen to that extent in healthy individuals. To our knowledge, such a lesion has never been reported in a case of animal EB, and these are findings seen more often in the severe generalized than intermediate variants of human JEB; they are typically not found in localized JEB [26].

In this study, we identified a homozygous missense variant, *LAMB3*:p.Cys392Arg, as a candidate causative variant for a new JEB in Australian Shepherd dogs. *LAMB3* encodes the laminin β3 chain, which, together with the α3 and γ2 chains, forms the heterotrimer laminin-332. Laminin β3 has two coiled-coil domains for the heterotrimer formation with the α3 and γ2 chains at its C-terminal end. The N-terminus consists of a globular domain (LN) and six laminin-type epidermal growth factor-like (LE) repeats [8,22]. The LE domains have conserved disulfide bonds, which may be important for the tertiary structure of these domains [27,28]. The LN and LE domains form a short arm in the cross-shaped laminin-332 heterotrimer, and mediate binding to type VII collagen in hemidesmosomes, which are necessary for the stable association between the epithelium and the stroma underneath [8,11].

The p.Cys392Arg variant changes one of the highly conserved cysteine residues in the third LE domain, which prevents the formation of the disulfide bond between Cys-392 and Cys-379. We hypothesize that this may lead to a change in the tertiary structure of laminin β3, and impair the binding of laminin-322 to collagen type VII in hemidesmosomes. Further experiments at the protein level are required in order to confirm this putative pathomechanism.

With this description, we now have two variants of canine intermediate JEB due to variants in related genes (*LAMA3* in German Shorthaired Pointers and *LAMB3* in Australian Shepherds) encoding the laminin α3 and β3 chains that assemble with the γ2 chain to form the laminin-332 heterotrimer. In both of these breeds, the variants are predicted to result in some residual protein function (Australian Shepherds), or in the secretion of some normal laminin-332 trimers (German Shorthaired Pointers) [5], which may explain the similar absence of lesions at birth and the intermediate clinical phenotype.

In humans, the specific variants and their consequences at the mRNA and protein levels contribute to the spectrum of severity encountered in different subtypes of EB [10]. Severe forms of JEB are associated with nonsense, frameshift, or out-of-frame splicing variants that result in nonfunctional or complete loss of the protein. Intermediate JEB occurs when a laminin chain is mutated, but the LM-332 heterotrimer can still form, which is often the case for missense variants [16]. Missense variants affecting cysteine residues in the LE domains, *LAMB3*:p.Cys355Arg and p.Cys433Trp, have been reported in human patients with intermediate JEB [16,17,18,19]. The clinical phenotype observed in the investigated dogs homozygous for p.Cys392Arg can also be classified as JEB of intermediate severity, and corresponds well to the human spectrum of genotype–phenotype correlations.

## 5. Conclusions

We characterized a new recessive form of JEB in Australian Shepherd dogs. A precision medicine approach identified a missense variant in the *LAMB3* gene, c.1174T>C or p.Cys392Arg as likely candidate causative variant. Our data enable genetic testing to avoid the unintentional breeding of further affected dogs and provide the first spontaneous large animal model for JEB due to altered laminin β3.

## Figures and Tables

**Figure 1 genes-11-01055-f001:**
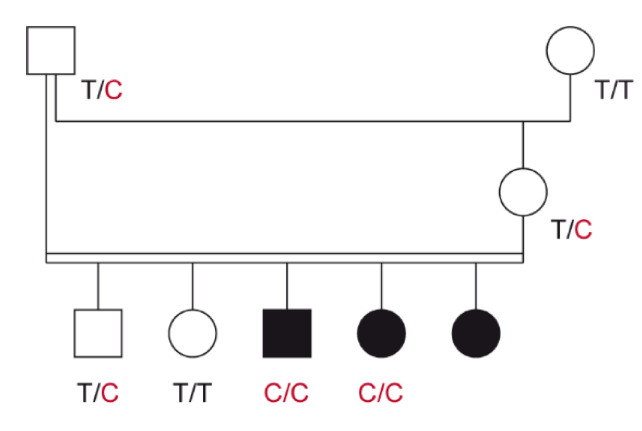
Pedigree of the investigated Australian Shepherd family. Squares represent males and circles represent females. The three affected puppies are indicated by the filled symbols. Note that the father of the litter was simultaneously the maternal grandfather. A close inbreeding loop greatly increases the risk for recessive hereditary defects. Genotypes at the *LAMB3*:c.1174T>C variant are indicated for all animals, from which a DNA sample is available (see Section 3.2).

**Figure 2 genes-11-01055-f002:**
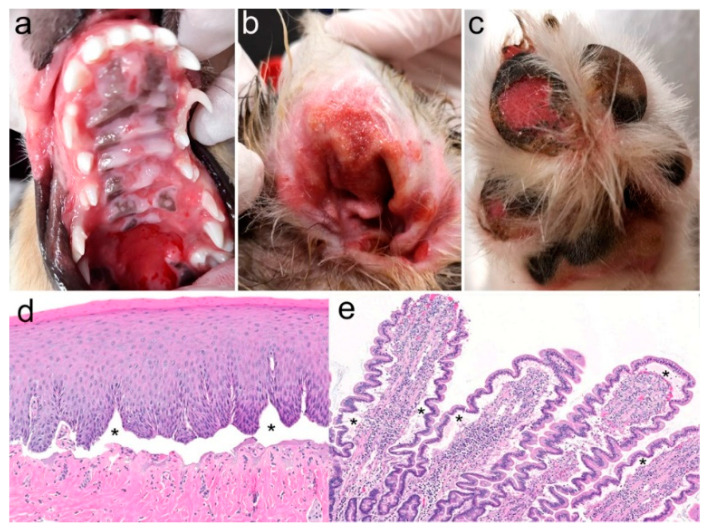
Clinical and histopathological phenotype. (**a**) Severe coalescing ulcers on the gingiva and hard and soft palate, (**b**) concave pinna (**c**) and footpads. Biopsy samples collected from the (**d**) oral cavity and (**e**) duodenum revealed widespread separation of the epithelium from the underlying connective tissue (asterisks).

**Figure 3 genes-11-01055-f003:**
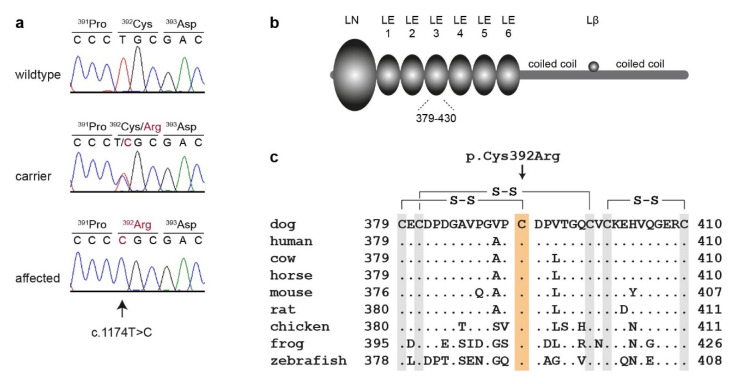
Details of the *LAMB3*:c.1174T>C, p.Cys392Arg variant. (**a**) Representative electropherograms of three dogs with different genotypes are shown. The variable position is indicated by an arrow, and the amino acid translations are shown. (**b**) Domain organization of the 1172 amino acid laminin β3 precursor [8]. The N-terminus consists of a globular domain (LN), followed by six laminin EGF-like (LE) domains. These N-terminal domains are located in the basement membrane and may be involved in binding to collagen VII. The C-terminal half of laminin β3 participates in two coiled-coil domains that mediate trimerization with the α3 and γ2 chains in the laminin-332 heterotrimer. The small Lβ domain mediates the binding of agrin. (**c**) Multiple-species alignment of the beginning of the LE3 domain harboring the p.Cys392Arg variant. The variant affects a highly conserved cysteine residue that forms a disulfide bridge with Cys-379 [22]. Note that all six cysteine residues in this region contribute to disulfide bonds, and are strictly conserved across vertebrates.

**Table 1 genes-11-01055-t001:** Consensus reclassification of epidermolysis bullosa and other disorders with epidermal fragility and their known functional candidate genes, as of 2020 [1].

Disorder	Level of Cleavage	Gene	Protein	Inheritance ^1^
**Classical Epidermolysis Bullosa (EB)**				
EB simplex (EBS)	Basal epidermal	*CD151*	CD151 molecule (Raph blood group)	AR
		*DST*	dystonin	AR
		*EXPH5*	exophilin 5	AR
		*KLHL24*	kelch like family member 24	AD
		*KRT5*	keratin 5	AD, AR
		*KRT14*	keratin 14	AD, AR
		*PLEC*	plectin	AR
Junctional EB (JEB)	Junctional	*COL17A1*	collagen type XVII, α 1 chain	AR
		*ITGA3*	integrin subunit α 3	AR
		*ITGA6*	integrin subunit α 6	AR
		*ITGB4*	integrin subunit β 4	AR
		*LAMA3*	laminin subunit α 3	AR
		*LAMB3*	laminin subunit β 3	AR
		*LAMC2*	laminin subunit γ 2	AR
Dystrophic EB (DEB)	Dermal	*COL7A1*	collagen type VII, α 1 chain	AD, AR
Kindler EB	Mixed	*FERMT1*	fermitin family homolog 1	AR
**Other Disorders with Skin Fragility**				
Peeling skin disorders	Intraepidermal	*CAST*	calpastatin	AR
		*CSTA*	cystatin A	AR
		*CTSB*	cystatin B	AR
		*DSG1*	desmoglein 1	AR
		*FLG2*	filaggrin family member 2	AR
		*SERPINB8*	serpin family B member 8	AR
		*SPINK5*	serine peptidase inhibitor Kazal type 5	AR
Erosive skin fragility disorders	Intraepidermal	*DSC3*	desmocollin 3	AR
		*DSG3*	desmoglein 3	AR
		*DSP*	desmoplakin	AR
		*JUP*	junction plakoglobin	AR
		*PKP1*	plakophilin 1	AR
Keratinopathic ichthyoses	Intraepidermal	*KRT1*	keratin 1	AD
		*KRT2*	keratin 2	AD
		*KRT10*	keratin 10	AD, AR
Pachyonychia congenita	Intraepidermal	*KRT6A*	keratin 6A	AD
		*KRT6B*	keratin 6B	AD
		*KRT6C*	keratin 6C	AD
		*KRT16*	keratin 16	AD
		*KRT17*	keratin 17	AD
Syndromic connective tissue disorder with skin fragility	Dermal	*PLOD3*	procollagen-lysine,2-oxoglutarate 5-dioxygenase 3	AR

^1^ AD—autosomal dominant; AR—autosomal recessive.

**Table 2 genes-11-01055-t002:** Results of variant filtering in the affected Australian Shepherd dog against 73 control genomes. Only homozygous variants are reported.

Filtering Step	Variants
All variants in the affected dog	3,111,811
Private variants	11,754
Protein-changing private variants	54
Protein-changing private variants in 37 candidate genes	1

**Table 3 genes-11-01055-t003:** Genotype-phenotype association of the *LAMB3*:c.1174T>C variant with JEB.

Dogs	T/T	T/C	C/C
Cases (*n* = 2) ^1^	-	-	2
Controls, Australian Shepherd dogs (*n* = 245)	242	3	-
Controls, other breeds (*n* = 663) ^1^	663	-	-

^1^ These genotypes were derived from 590 genome sequences reported in the literature [20], and the 73 control genomes used in this study.

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
