# Peer review of "LAMB3 Missense Variant in Australian Shepherd Dogs with Junctional Epidermolysis Bullosa"

_genes, 2020, doi:10.3390/genes11091055_

Round 1

Reviewer 1 Report

The authors describe a junction epidermolysis bullosa in three Australian shepherd dogs from one litter and the identification of a mutation in LAMB3gene that may be implicated in the disease. The introduction is a clear review of the current classification of the skin fragility disorders, both in human and in domestic animals.

Overall the paper is showing clean data and results.

I only have two minor comments:

The candidate variant is quite strong, as it segregates perfectly in the breed, is not found in other dog breeds, and is located in a gene already known in such skin condition in human. However, even if I am convinced by all these genetic data, I would have like to have few functional data to comfort the hypothesis stated, page 8, lines 278-280, that the mutation impairs “the binding of laminin-322 to collagen type VII in hemidesmosomes”. If such data already exist in other models the authors should at least discuss this part in few lines.

My other comment is on Table S3 concerning the private variants of the sequenced dog. As there are 33727 lines in the table, it is quite hard to find the LAMB3 variant, even if it is highlighted in yellow, line 8505… I suggest to add a second sheet to this excel file, if this is allowed by the journal, which will only contains the 54 protein-changing private variants. This will help readers to rapidly see the genes that contain such variants and the type of variants: when doing this filtering rapidly, I counted 49 missense variants, so what are the other 5?

Author Response

(1)

The candidate variant is quite strong, as it segregates perfectly in the breed, is not found in other dog breeds, and is located in a gene already known in such skin condition in human. However, even if I am convinced by all these genetic data, I would have like to have few functional data to comfort the hypothesis stated, page 8, lines 278-280, that the mutation impairs “the binding of laminin-322 to collagen type VII in hemidesmosomes”. If such data already exist in other models the authors should at least discuss this part in few lines.

Response: We appreciate this comment by the reviewer. Unfortunately, to the best of our knowledge, only the physiological binding of the N-terminus of laminin β3 to the vWFA2 subdomain of collagen VII has been experimentally confirmed (stated in lines 272-275 of the revised manuscript). We are not aware of any studies demonstrating impaired binding of laminin β3 to collagen VII in LAMB3 mutants. To address the comment, we slightly revised our statement and emphasized that this is an unproven hypothesis requiring further experimental validation at the protein level.

(2)

My other comment is on Table S3 concerning the private variants of the sequenced dog. As there are 33727 lines in the table, it is quite hard to find the LAMB3 variant, even if it is highlighted in yellow, line 8505… I suggest to add a second sheet to this excel file, if this is allowed by the journal, which will only contains the 54 protein-changing private variants. This will help readers to rapidly see the genes that contain such variants and the type of variants: when doing this filtering rapidly, I counted 49 missense variants, so what are the other 5?

Response: We revised TableS3 accordingly and added a second sheet listing only the protein-changing variants. According to our counting, there are indeed 54 protein-changing variants comprising 50 variants with moderate impact (49 missense, 1 in-frame deletion), and 4 variants with high impact (2 splice variants, 1 stop lost, 1 frameshift insertion).

Reviewer 2 Report

I just want to call your attention on one points:

-Would you mind verifying the reference number 19?. In case a mistake exist in this reference, perhaps the correction of the subsequent references numbering should be checked (as it is well known, this is the main problem of this system).

Author Response

Would you mind verifying the reference number 19? In case a mistake exist in this reference, perhaps the correction of the subsequent references numbering should be checked (as it is well known, this is the main problem of this system).

Response: Thank you for spotting the incorrect citation. Fortunately, there was only one instance in line 120, which we revised accordingly. All other citations should be correct.

Reviewer 3 Report

The authors clearly demonstrate a missense variant in LAMB3 as a causative variant for junctional epidermolysis bullosa in Australian Shepherd. The authors identify the variant, describe the effect of an amino acid change in the biochemical behavior of the protein, and then relate this back to a clinical diagnosis. The paper is structured, organized, easy to read and scientifically sound. I have no concerns with the paper or methodology.

Author Response

The authors clearly demonstrate a missense variant in LAMB3 as a causative variant for junctional epidermolysis bullosa in Australian Shepherd. The authors identify the variant, describe the effect of an amino acid change in the biochemical behavior of the protein, and then relate this back to a clinical diagnosis. The paper is structured, organized, easy to read and scientifically sound. I have no concerns with the paper or methodology.

Response: Thank you for the positive comments.